biophysics/cellular biology/computational biology

Ising model, phase transition, bystander effect, cancer cell response, cytotoxicity

**Author for correspondence:**
J. A. Tuszynski
e-mail: jack.tuszynski@gmail.com

# Cell death and survival due to cytotoxic exposure modelled as a two-state Ising system

S. Arbabi Moghadam[1], V. Rezania[2] and J. A. Tuszynski[1,3,4]

[1]Department of Physics, University of Alberta, Edmonton, Alberta, Canada T6G 2E1
[2]Department of Physical Sciences, MacEwan University, Edmonton, Alberta, Canada T5 J 4S2
[3]Department of Oncology, University of Alberta, Edmonton, Alberta, Canada T6G 1Z2
[4]DIMEAS, Politecnico di Torino, Turin, Italy

Cancer chemotherapy agents are assessed for their therapeutic utility primarily by their ability to cause apoptosis of cancer cells and their potency is given by an IC50 value. Chemotherapy uses both target-specific and systemic-action drugs and drug combinations to treat cancer. It is important to judiciously choose a drug type, its dosage and schedule for optimized drug selection and administration. Consequently, the precise mathematical formulation of cancer cells' response to chemotherapy may assist in the selection process. In this paper, we propose a mathematical description of the cancer cell response to chemotherapeutic agent exposure based on a time-tested physical model of two-state multiple-component systems near criticality. We describe the Ising model methodology and apply it to a diverse panel of cytotoxic drugs administered against numerous cancer cell lines in a dose–response manner. The analysed dataset was generated by the Netherlands Translational Research Center B.V. (Oncolines). This approach allows for an accurate and consistent analysis of cytotoxic agents' effects on cancer cell lines and reveals the presence or absence of the bystander effect through the interaction constant. By calculating the susceptibility function, we see the value of IC50 coinciding with the peak of this measure of the system's sensitivity to external perturbations.

## 1. Introduction

Chemotherapy is a standard cancer therapy modality based on the concept of cytotoxicity of drugs or drug combinations inflicting lethal damage to cancer cells but being less damaging to normal cells. Cytotoxicity refers to killing or damaging a viable cell by chemical compounds or pathogens. Toxic agents usually damage molecular targets such as metabolic sites, signalling proteins or DNA, which are essential to the cell's reproductive ability and

survival. The result is dose-dependent cell death or inhibition of its proliferative potential. Unlike radiotherapy, which specifically targets cancer cells in a tumour, chemotherapy is typically applied systemically and is not site-specific unless combined with target-specific antibodies or via special drug delivery strategies. Hence, it may also affect metastasized cells distant from the primary tumour site and also the entire body of the patient with concomitant detrimental side effects [1–4]. Besides the collateral damage to healthy cells, another major shortcoming of chemotherapy is the emergence of drug resistance in the population of tumour cells. This is commonly due to the heterogeneity of tumour cells, some of which are sensitive to a given drug and others resist it. Due to the survival of the fittest, the resistant subpopulation proliferates when exposed to a cytotoxic agent, while sensitive subpopulation is eradicated making the tumour more malignant over time. Consequently, we should judiciously choose a pharmacological agent, its dose and scheduling, which can be a very complex, multi-factorial problem, considering both tumour destruction and the collateral damage to the healthy tissues. Furthermore, chemotherapy-transfected cells are likely to host toxic anabolites resulting from the therapy, which can directly transfer into neighbouring untransfected cells through diffusion resulting in a secondary wave of damaged cells. This refers to a so-called bystander effect extensively reported in the literature [5–15]. In general, it describes the population of dead and/or damaged cells that are not directly targeted by either chemotherapy or irradiation.

Cytotoxicity mechanisms are commonly analysed using the Hill model representing an empirical sigmoidal fit describing the binding equilibria in ligand–receptor interactions [16] based on a simple reaction scheme: $R + nL \overset{K_d}{\leftrightarrow} RL_n$, where $R$ is the receptor, $n$ (called the Hill coefficient) is the number of ligands $L$ and $K_d$ is the dissociation constant. Despite its simplicity, the Hill equation is not always physically plausible and the Hill coefficient $n$ can only be accurately estimated for extremely positive cooperative interactions among multiple ligand-binding sites. Even for a reaction with a high degree of positive cooperativity, e.g. binding four oxygen molecules to haemoglobin, the Hill coefficient ranges from 1.7 to 3.2 rather than 4 [16]. Therefore, other physically plausible reaction schemes such as a 'two-state' (activated and inactivated) receptor model have been proposed to account for complex cases with various ligand–receptor cooperativities [16]. The main limitation of such models is a large number of adjustable parameters required to fit experimental data, e.g. seven parameters are needed for the haemoglobin-oxygen two-state receptor model. In addition, the observed bystander effects cannot be addressed by the above models. For these reasons, we propose a more accurate and better-motivated modelling approach to cytotoxicity, following on its successful applications in physics and other fields. To improve a statistical analysis of the effects of chemotherapeutic agents on tumour cells, we adopt the concepts introduced for phase transitions and multi-stability. This is appropriate for cells under cytotoxic attack since cytotoxicity (and irradiation) is a dynamical process, which triggers a stochastic transition from a proliferating cell (live) to a non-proliferating cell (dead or senescent). As a result, cytotoxicity can be viewed as a transition between two different biological states present in replicas of manifestly identical systems (cancer cell cultures). Phase transitions have been successfully analysed in many-body systems in physics, chemistry and even social sciences and economics. Phase transitions ranging from the solid ↔ liquid ↔ gas transitions at the macroscopic level to the superconductor-metal transition at the microscopic level have been exquisitely understood employing statistical model systems such as the Ising or the Landau model (see electronic supplementary material, Text for details). Bistability is a common motif in systems undergoing phase transitions, which can exist in two distinct states and switch between them at the transition point in response to a change in the so-called control parameters [17]. The system's response is reflected in its order parameter, i.e. a macroscopic property that is zero in the disordered phase and non-zero in the ordered phase. The so-called generalized susceptibility function is the first-order derivative of the order parameter with respect to the control parameter, and it describes the system's sensitivity to perturbations. Generalized susceptibility of an infinitely large system diverges at the critical point (the tipping point) as the system switches from one stable phase to the other. Biological systems such as cancer cells positioned at a threshold of viability can indeed be viewed as dynamical systems at criticality, which exhibit clear bistability characteristics between being alive and dead. This perspective leads to a meaningful connection between biology and physics providing a physical model with a better mathematical insight into cancer cells' behaviour [17–21].

Our aim is to implement the spin-1/2 Ising model of phase transitions as an elegant and powerful mathematical approach to study cancer cells exposed to cytotoxic chemotherapeutic agents (see electronic supplementary material, Text for details) [17,22–41]. Recently, a similar approach has been applied to ionizing radiation response of tumours [5]. In analogy to the Ising model for ferromagnetic materials with long-range interactions, these authors proposed to study tumour response to a uniform ionizing radiation field. In particular, using the mean-field approach, individual cells are averaged out

and characteristic features such as cell survival curves, tumour control probabilities, fractionation and bystander effects emerge naturally showing that the bystander effects cannot be ignored at low-dose radiotherapy [5]. Below, we elaborate on the use of the Ising model to a dose–response cancer cell survival dataset provided by the Netherlands Translational Research Center B.V. (Oncolines). These experimental data correspond to inhibition profiles obtained in a uniform manner for numerous cell lines exposed to a number of chemotherapy agents [42,43].

## 2. Methods

Similarly to the Ising model with two spin states, up and down, the effect of chemotherapeutic drugs on cancer cells is associated with two possible outcomes: either survival or death states of cancer cells, respectively. For simplicity, we assign two states as $s_i = \{\pm 1\}$, where $s_i = 1$ refers to the live state of the $i$th cell and $s_i = -1$ represents the dead state of the $i$th cell. Following the radiation-induced bystander effect [5,6], we invoke the bystander effect for chemotherapy-exposed cancer cells in a culture by introducing a classical Ising Hamiltonian applicable for interacting spin systems as

$$\mathcal{H} = -\sum_{ij}^{N} J_{ij} s_i s_j + \sum_{i=1}^{N} h_i (1 - 2s_i) , \qquad (2.1)$$

where $ij$ indices are summed over all nearest-neighbour cell pairs at site $i$ and $j$, $J_{ij}$ denotes the strength of the interaction between neighbouring cells $i$ and $j$ (namely the strength of the bystander effect) and $h_i$ represents the potency of the external agent at location of $s_i$. $J_{ij} = 0$ means there is no interaction between cells. Similar to spin systems, it is expected that the interaction strength is always positive for all cells $i$ and $j$, so $J_{ij} \geq 0$, and the summation $\sum_{ij}^{N} J_{ij}$ over all the neighbouring cells is finite. Using statistical mechanic approaches and mean-field approximation for interacting and non-interacting cells, the partition function, $Z$, and the order parameter, $M$, are found after several steps of calculations as (see electronic supplementary material, Text for calculation details)

$$Z = \left[ 2e^{-\lambda M(M-1)} \cosh \left( \frac{h}{k_B T} + \lambda M \right) \right]^N \qquad (2.2)$$

and

$$2M - 1 = \tanh \left( \frac{h}{k_B T} + \lambda M \right) \qquad (2.3)$$

where $\lambda = J/2k_B T$ [5]. Using the fact that the order parameter, $M$, and control parameter, $h$, are equivalent in the case of cancer cells to the death rate, $R$, and the logarithm of concentration, $\log(C)$, respectively, we rewrite equation (2.3) as

$$2R - 1 = \tanh \left[ \frac{h}{k_B T} + \lambda R \right] . \qquad (2.4)$$

Inserting electronic supplementary material equation (S9) in equation (2.4), we find that

$$R = \frac{1}{2} \left( 1 + \tanh \left[ 1.15\alpha \log \left( \frac{C}{C_M} \right) + \lambda R \right] \right) , \qquad (2.5)$$

where $1.15 = \ln(10)/2$, $C$ is the drug concentration and $C_M$ is a Michaelis constant representing the drug concentration associated with reaching the half-maximal inhibition effect [44]. The parameter $\alpha$ shows the slope of the dose–response curve and depends on the system's heterogeneity, drug-binding efficacy and diffusion of drug molecules. It can be estimated by fitting the solution to experimental data points. In cytotoxicity assays, $C_M$ is usually denoted as IC50. As shown in the next section, equation (2.5) provides excellent agreement with the cytotoxicity experiment data. Equation (2.5) has a critical value for $\lambda$, at which a discontinuity in the death rate emerges and can be solved numerically for given values of $\alpha$ and $\lambda$. Figure 1$a$–$c$ illustrates the death rate, $R$, as a function of $\log(C/C_M)$ for four values of $\alpha$ and $\lambda$, respectively. Figure 1$a$ shows the result for uncorrelated cells, which presents similar behaviour to that found in the Landau theory of phase transition (see electronic supplementary material, Text, appendix and figure S1 for details) [33–35,38]. As shown in figure 1$a$–$c$, increasing the interaction strengths between cells for $\lambda = 0$, 0.5, and 1 triggers their transit from live to dead states as a result of only a subtle change in the drug concentration. As stated above, the generalized

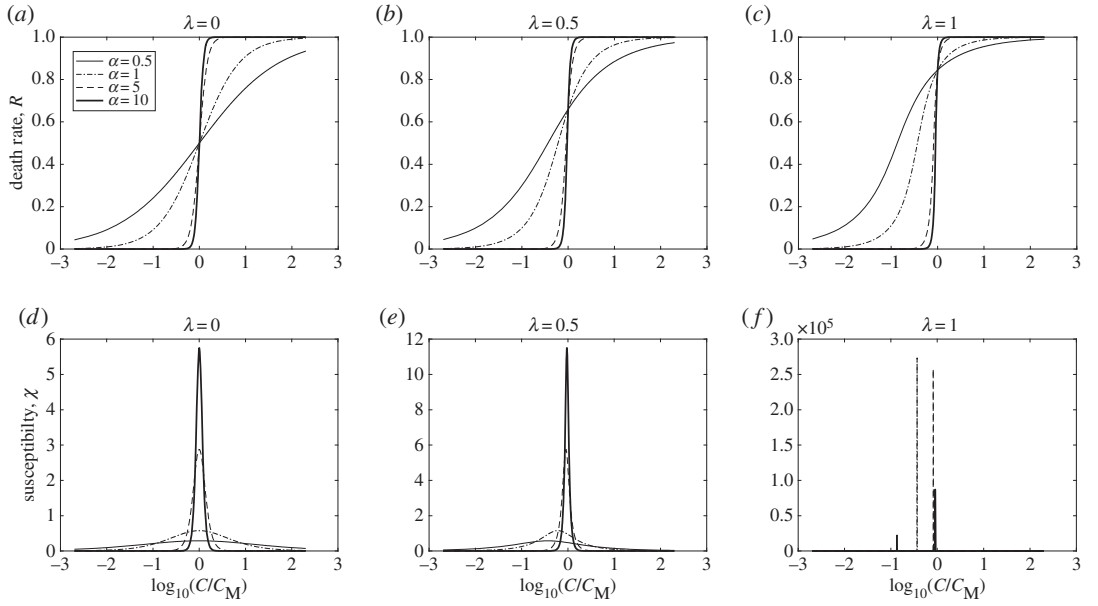

**Figure 1.** Solution for death rate (a–c) and susceptibility (d–f) of equations (2.5) and (2.6), for four values of $\alpha = 0.5$, 1, 5 and 10; (a) and (d) $\lambda = 0$, (b) and (e) $\lambda = 0.5$, (c) and (f) $\lambda = 1$.

susceptibility function, $\chi = \partial M / \partial h$, describes the sensitivity of the order parameter to a change in the control parameter [45,46], which here corresponds to the survival rate's change due to the change in the drug concentration. Using the Ising model results, equation (2.5), the cancer cell susceptibility function is calculated as the first derivative of the death rate, $R$, with respect to the $\log(C/C_M)$

$$\chi = \frac{1}{2} \left[ \frac{1.15\alpha(1 - \tanh^2\delta)}{1 - \frac{\lambda}{2}(1 - \tanh^2\delta)} \right], \tag{2.6}$$

where $\delta = 1.15\alpha \log(C/C_M) + \lambda R$. In the case of no interaction, $\lambda = 0$, using equation (2.6), we show that the susceptibility function maximum occurs at 'zero-field' or $C = C_M$ with $\chi_{\max} = 0.575\alpha$. The susceptibility of the Ising model was extensively studied by Fisher [45,46]. Lower panels in figure 1 depict susceptibility calculated using equation (2.6) as a function of $\log(C/C_M)$ for various values of $\alpha$ and $\lambda$, respectively. In the case $\lambda = 0$, figure 1d, the susceptibility maximum occurs at zero-field, while by increasing $\lambda$ it peaks at concentrations below $C_M$. Interestingly, by increasing $\alpha$ in the non-zero $\lambda$ case, the maximum shifts towards zero-field. This demonstrates that there is a competition between $\alpha$ and $\lambda$, where larger values of $\alpha$ provide solutions similar to $\lambda = 0$. This is observed in the parameter values found using the experimental data.

# 3. Results and discussion

In this study, we applied the Ising model methodology to better understand and more accurately describe cancer cell response to chemotherapy agents. Inhibition profiles of 13 diverse anti-cancer compounds were analysed from proliferation assays performed on 66 cancer cell lines provided by Oncolines, Inc., The Netherlands [42,43]. The anti-cancer compounds tested were Afatinib, Bortezomib, Busulfan, Cisplatin, Doxorubicine, Idelalisib, Irinotecan, Methotrexate, Paclitaxel, Palbociclib, Tazemetostat, Trametinib and Vincristine [47–51] (see electronic supplementary material, Text, appendix and table S1 for details). For all the 13 compounds and 66 cell lines, the death rate, $R$, is plotted in terms of the logarithm of anti-cancer drug concentration, $\log(C)$, and fitted to the following function:

$$R = a[b + \tanh(c(\log(C) + d))], \tag{3.1}$$

where $a$, $b$, $c$ and $d$ are the best-fit parameters. As an example, figure 2a shows the results for Bortezomib acting on a melanoma cell line (A375). Based on the cell response-Ising model, the IC50 value marks the drug concentration at which the phase transition from a live system to a dead one occurs. In figure 2a,

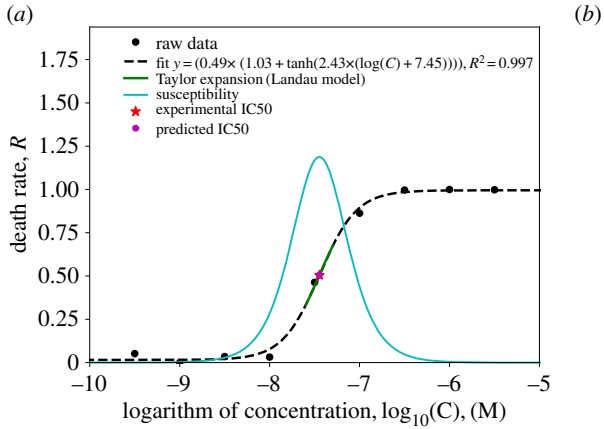
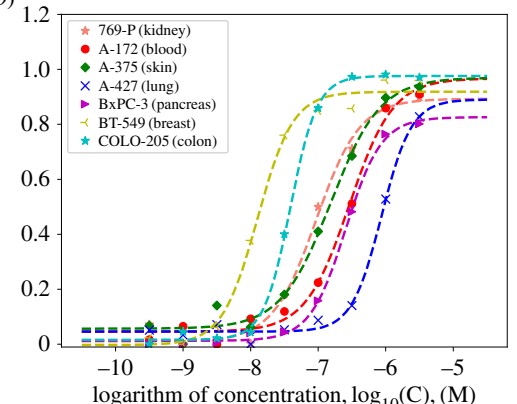

**Figure 2.** (a) Profile of the death rate in terms of the logarithm of the drug concentration for the melanoma A375 cell line in the exposure of Bortezomib drug. Dashed line represents the fit to equation (3.1), the solid line in cyan is the corresponding susceptibility, the red star represents the experimental IC50, the purple circle shows the predicted IC50 value from our fitting function and the solid green line shows the Taylor expansion around the coefficient d. (b) Profile of the death rate in terms of the log(C) of Paclitaxel drug for different cell lines.

dashed lines represent the best-fitting function and a red star and a purple circle represent the experimental and predicted IC50 values, respectively. Taylor expansion of the fitted function around IC50 is shown using a green solid line in figure 2a. Note that the solid line in cyan shows the susceptibility in equation (2.6) and its highest value coincides with the IC50 concentration, which provides a rationale for the thus far arbitrary use of IC50 as a significant parameter for cytotoxicity estimates. These findings also demonstrate good agreement between our model and the Landau mean-field theory of phase transition (see electronic supplementary material, Text). Figure 2b illustrates a similar behaviour for Paclitaxel applied to other cell lines, namely: 769-P (kidney), A-172 (blood), A-375 (skin), A-427 (lung), BxPC-3 (pancreas), BT-549 (breast) and Colo-205 (colon). All findings in this section demonstrate good agreement between the proposed model and the experimental data.

One of this model's advantages is its ability to predict a precise value of IC50 by finding the parameter $d$ of the fitting function. For those cytotoxic drugs, which follow the model, the difference between the predicted and experimental values of IC50 is very minor, of the order of $10^{-3}$. Comparing the fitted curve with equation (2.5) derived from the cell response-Ising model in table 1, one finds excellent agreement between the predicted and observed results for non-interacting cell lines ($\lambda = 0$ in equation (2.5)). Table 1 shows the average value of the fitting parameter for all the 66 cell lines tested followed by the correlation coefficient for each cytotoxic drug between the experiment and the cell response-Ising model. The best-fit parameter values among the cell lines are fairly consistent for $a$, $b$, $R^2$ and $\chi$ among the cytotoxic drugs, although some small fluctuations can be seen for parameter $c$ and $d$ values. To study the possibility of the bystander effect, we use equation (2.5), derived using the Ising model, with an assumption that $\alpha = 1$. The obtained values are fairly consistent among all the drugs (see electronic supplementary material, appendix and figure S3). It emerges that drugs with higher correlations to our model have high susceptibility values as well. The correlation coefficient includes the cell lines with $R^2 > 0.5$ and the susceptibility $\chi < 10$. These two conditions drop approximately 16% of the cases studied, which do not follow the model. One possible reason for this is that the experimental IC50 reported is not appropriate, possibly meaning that the drug was not cytotoxic for that particular cell line, or the drug dosage was not sufficiently high.

In addition, table 2 lists the fitting results of our model to all the cytotoxic drugs for interacting cells, $\lambda \neq 0$, and we fitted the data with the equation $R = 0.5(1 + \tanh(1.15(\log(C) - \log(\text{IC50})) + \lambda R)$. Although for some drugs such as Trametinib, Idelalislib, Tazemetostat and Busulfan, only a few of the cell lines have been fitted very well to the model; the results show the existence of the bystander effect in most cell lines. It is important to note that no obvious trend has been found between the interacting and non-interacting cell lines for each drug. In other words, the cell lines that were not following the model in the non-interacting case are not well fitted to the model in the interacting case either.

In the original dataset, there are some cell lines for which the corresponding IC50 value was not reported. This is due to it exceeding the maximum tested concentration (less than 31 600 nM). At this concentration, the compounds still show less than 50% inhibition of cell proliferation. Figure 3 shows the

**Table 1.** The best-fit parameters to equation (3.1), $a$, $b$, $c$ and $d$, susceptibility, $\chi$ and correlations with the theoretical model (± stands for standard deviation).

| predicted parameters based on the Ising model | | | | | | | |
|---|---|---|---|---|---|---|---|
| cytotoxic drugs | $a$ | $b$ | $c$ | $d$ | | | |
| equation (3.1) | 0.5 | 1 | $1.15\alpha$ | $-\log_{10}(IC50)$ | | | |
| estimated fitting parameters of equation (3.1) when $\lambda = 0$ | | | | | | | |
| cytotoxic drugs | $a$ | $b$ | $c$ | $d$ | $R^2$ | $\chi$ | total correlation (%) |
| Bortezomib | 0.47 ± 0.02 | 1.08 ± 0.06 | 5.98 ± 6.49 | 7.68 ± 0.36 | 0.99 ± 0.01 | 2.57 ± 2.7 | 100.0 |
| Methotrexate | 0.45 ± 0.06 | 1.09 ± 0.08 | 8.16 ± 7.54 | 7.66 ± 0.35 | 0.99 ± 0.01 | 2.99 ± 2.6 | 100.0 |
| Paclitaxel | 0.43 ± 0.07 | 1.07 ± 0.07 | 2.37 ± 1.01 | 7.52 ± 0.55 | 0.99 ± 0.01 | 1.01 ± 0.5 | 100.0 |
| Vincristine | 0.45 ± 0.04 | 1.09 ± 0.07 | 4.37 ± 4.14 | 8.00 ± 0.61 | 0.99 ± 0.02 | 1.80 ± 1.4 | 100.0 |
| Doxorubicin | 0.48 ± 0.06 | 1.07 ± 0.09 | 2.16 ± 2.60 | 6.96 ± 0.49 | 0.99 ± 0.03 | 0.99 ± 1.1 | 97.0 |
| Irinotecan | 0.43 ± 0.11 | 1.14 ± 0.12 | 3.37 ± 4.72 | 5.51 ± 0.54 | 0.97 ± 0.04 | 1.21 ± 1.5 | 95.5 |
| Cisplatin | 0.54 ± 0.26 | 1.10 ± 0.06 | 2.37 ± 3.45 | 5.16 ± 0.54 | 0.98 ± 0.02 | 1.11 ± 1.4 | 93.9 |
| Afatinib | 0.51 ± 0.09 | 1.14 ± 0.16 | 2.38 ± 1.25 | 5.47 ± 0.37 | 0.98 ± 0.02 | 1.15 ± 0.6 | 87.7 |
| Trametinib | 0.31 ± 0.20 | 1.15 ± 0.36 | 3.92 ± 6.63 | 7.69 ± 0.90 | 0.92 ± 0.11 | 0.60 ± 0.7 | 72.7 |
| Idelalisib | 0.67 ± 0.75 | 1.16 ± 0.20 | 3.95 ± 6.09 | 4.43 ± 0.88 | 0.91 ± 0.10 | 1.00 ± 1.1 | 71.2 |
| Tazemetostat | 0.20 ± 0.17 | 1.51 ± 0.45 | 6.50 ± 10.57 | 5.18 ± 0.71 | 0.80 ± 0.16 | 1.32 ± 1.6 | 62.1 |
| Palbociclib | 0.63 ± 0.27 | 1.10 ± 0.10 | 2.95 ± 4.84 | 5.14 ± 0.48 | 0.98 ± 0.02 | 1.50 ± 2.2 | 59.1 |
| Busulfan | 0.52 ± 1.65 | 1.44 ± 0.45 | 10.12 ± 9.13 | 4.49 ± 3.50 | 0.70 ± 0.17 | 1.76 ± 2.3 | 53.0 |

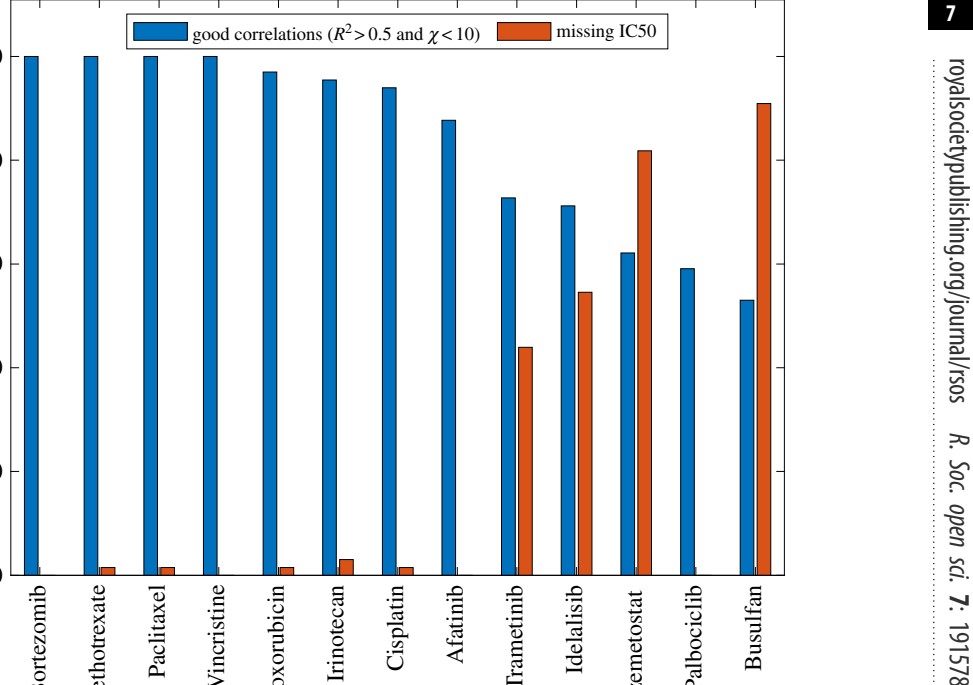

**Figure 3.** Correlations with the cell response-Ising model and the corresponding IC50 values, blue bars represent the good correlations (%) and the red bars shows the missing IC50 reported for each cytotoxic drug (%).

**Table 2.** Ising model for interacting cells (± stands for standard deviation).

| cytotoxic drugs | $\lambda$ | $R^2$ | total correlation (%) |
| --- | --- | --- | --- |
| Bortezomib | 0.41 ± 0.21 | 0.96 ± 0.03 | 98.5 |
| Methotraxate | 1.01 ± 1.53 | 0.95 ± 0.04 | 75.8 |
| Afatinib | 0.98 ± 0.79 | 0.93 ± 0.09 | 66.7 |
| Vincristine | 1.18 ± 0.93 | 0.95 ± 0.08 | 63.6 |
| Doxorubicine | 1.00 ± 0.95 | 0.97 ± 0.03 | 57.6 |
| Paclitaxle | 1.22 ± 1.06 | 0.93 ± 0.10 | 56.1 |
| Cisplatin | 0.89 ± 0.72 | 0.96 ± 0.03 | 53.0 |
| Palboliclib | 0.71 ± 0.71 | 0.87 ± 0.20 | 51.5 |
| Irinotecan | 0.72 ± 0.34 | 0.80 ± 0.16 | 40.9 |
| Trametinib | 0.31 ± 0.24 | 0.90 ± 0.08 | 10.6 |
| Idelalilsib | 0.17 ± 0.04 | 0.81 ± 0.04 | 9.1 |
| Tazemetostat | 0.86 ± 0.59 | 0.83 ± 0.12 | 4.5 |
| Busulfan | 0.21 ± 0.12 | 0.89 ± 0.01 | 4.5 |

correlation of each cytotoxic drug's profile with the cell response-Ising model prediction and the percentage of cells with missing IC50 values. We found that Busulfan, Palbociclib, Tazemetostat, Idelalisib and Trametinib exhibit a relatively low correlation with the model, while these cytotoxic drugs, except Palbociclib, have a high number of missing IC50 values. On the other hand, cytotoxic drugs with perfect correlations have only one cell line without an IC50 value measured. Therefore, there appears to be an inverse relationship between the correlation coefficient and the missing IC50 values. Those cell lines, which are missing the IC50 values appear not to follow the Ising model. One conclusion that can be reached is that the Ising model as applied to cytotoxicity is sensitive to the IC50 values. For those cell

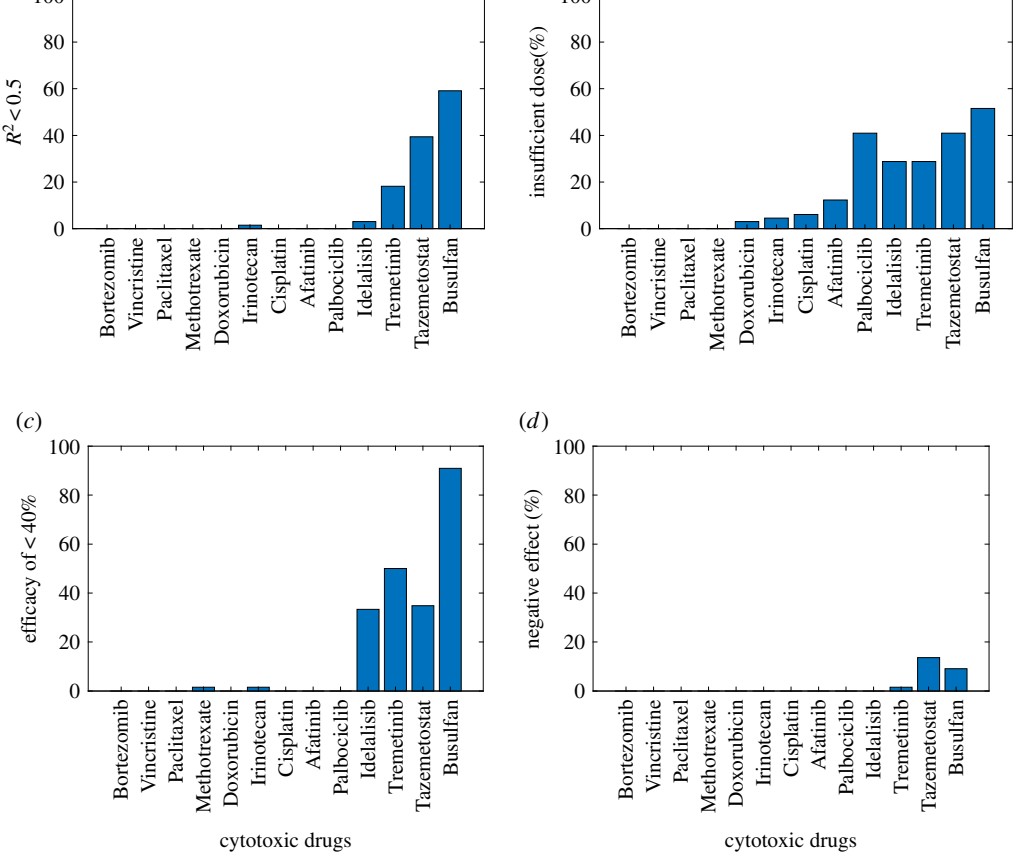

**Figure 4.** Weak correlation cell lines in different categories of (*a*) having $R^2 < 0.5$, (*b*) exhibiting insufficient cytotoxic dose, (*c*) showing reversal effect and (*d*) efficacy of less than 40% death rate.

lines for which the measurement has been made either below or above the IC50 values only, no phase transition has occurred so the model will not work (recall that the maximum value of the experimentally measured IC50 was 31.6 µM). Figure 3 shows correlation of the drugs with the cell response-Ising model in blue and the drugs with missing IC50 in red. Interestingly, eight of the drugs tested, namely Bortezomib, Methotrexate, Paclitaxel, Vincristine, Doxorubicin, Irinotecan, Cisplatin and Afatinib, exhibit excellent consistency with the cell response-Ising model (exceeding an 87% correlation coefficient), while for the rest of the drugs the correlation coefficient is between 52% and 73%. Since a diverse set of cancer cell lines was used among the 66 cell lines tested, this could have contributed to a low correlation coefficient for the drugs, which target a specific cancer type. For example, Busulfan is mostly used for treating bone marrow transplantation, especially in chronic myelogenous leukaemia (CML), such as represented by the following cell lines: SR, MOLT-4, K-562, SK-N-AS, SK-N-FI. The Ising model was fitted very well for bone marrow and CML cell lines in this case. This trend can also be seen for Trametinib, Idelalisib, Tazemetostat and Palbociclib. Correlated and uncorrelated cell lines have also been represented using green and red colours, respectively (see electronic supplementary material, Text, appendix and table S2).

Figure 4 shows the effect of cytotoxic drugs on the cell lines in different categories such as cell lines with $R^2 < 0.5$, exhibiting insufficient cytotoxic dose, showing a bizarre reversal effect (i.e. increasing the dose decreases the cells' death rate), showing efficacy of less than 40%. Figure 4*a* shows the number of the cells exposed to different drugs, which kill less than 40% of the cells. It demonstrates the cells that either were not responding well to the cytotoxic drugs or the drug was not cytotoxic enough. Comparing figure 4*a* and figure 3, one can see that the drugs with a missing IC50 value have a lower death rate, such as Busulfan, Tazemetostat, Trametinib and Idelalisib. Figure 4*b* illustrates the cell lines for which the dosage used was insufficient to cause the death of these cell lines. Also, figure 4*c* shows the efficacy of the drug on these cancer cell lines. Busulfan has lower efficacy in more than 90% of the cell lines. The efficacy value is 35% and 33% for Tazemetostat and Idelalisib, respectively, and 1.6% for Irinotecan and Methotrexate. Finally, figure 4*d* shows that some of the cell lines show a toxicity

**Figure 5.** Death rate profile for the colorectal cancer cell line, HCT116, exposed to Nitazoxanide and Mitomycin cytotoxic drugs (solid red line) [52]. Blue and black dashed lines represent the best-fit curves to the cell response-Ising model for interacting and non-interacting cases, respectively.

reversal effect when exposed to Tazemetostat, Busulfan and Trametinib such that increasing the cytotoxic dosage decreases the death rate, although the number of such cases is negligible.

As part of our study, the interacting cases with a non-zero $\lambda$ have also been considered. The same datasets have been fitted to equation (2.5) and the implicit equation for $\log(C)$ was solved. The interaction coefficients were assumed to be positive, $\lambda = J/2k_{B}T \geq 0$ in order to conform to the known biological effect. Refer to figure 1, in order to see the effect of the presence of the cell–cell interactions, the parameter $c$ of the fit should be less than 1 and $\lambda$ should be less than the critical value $\lambda < \lambda_{c} = 2$, at which fatal damage is created in the corresponding cells. The fitting procedure in the interacting case shows that the interaction parameter, $\lambda$, changes between 2.1 and 6.1 on average. In fact, since $\lambda$ is greater than $\lambda_{c}$, the cell–cell interactions are not significant here (table 2; electronic supplementary material, Text, appendix and figure S3).

Until now, all the cases analysed in this paper involved typical well-plate cytotoxicity assays, in which cell culture grows within a plane and is then exposed to toxic chemotherapy agents. Spatial dimensionality of physical systems undergoing phase transitions plays a crucial role in the response of the system to control parameter changes. In order to explore whether this can also be seen in biological systems such as cancer cells, we have found some experimental data in the literature that provide examples of dimensionality dependence. Figure 5 shows a comparison between the cell response-Ising model and the experimental data from reference [52] in which the drug Nitazoxanide was studied as a colorectal cancer therapy candidate. In fig. 2 of [52], HCT116 and HCT116 GPF cells were exposed to two cytotoxic drugs, Nitazoxanide and Mitomycin in monolayer two-dimensional and multi-cellular tumour spheroid (three-dimensional) cell cultures for 72 h. Here, dose–response curves for the two drugs have been compared to the cell response-Ising model in figure 5. The black dashed lines and blue dashed lines represent the best-fit values of the cell response-Ising model in the cases with a non-zero and zero $\lambda$, and the red line shows the HCT116 cell line exposed to Mitomycin (*a*) two-dimensional and (*b*) three-dimensional and Nitazoxanide (*c*) two-dimensional and (*d*) three-dimensional. It can be seen that in the case with no cell–cell interactions, $\lambda = 0$, the two cytotoxic drugs have a better correlation than in the interaction case. In

the presence of the interaction term, $\lambda \neq 0$, the values of $\lambda$ in the curves $a$ to $d$, are 1.32, $-3.01$, 0.74 and 0.05, respectively. These values show that the model does not work well for the Mitomycine three-dimensional experiment, while for Nitazoxanide three-dimensional, the $\lambda$ value shows that the cell–cell interactions are very weak (almost zero). However, in the case of the two-dimensional experiments for both drugs, we see a better correlation in the cell–cell interaction cases (figure 5$a$,$c$). We can, therefore, tentatively conclude that spatial dimensionality does indeed affect the response of cell cultures to cytotoxic agents but a more in-depth analysis of larger datasets is required to develop an appropriate mathematical model that captures these complex systems' behaviour better than the present Ising model.

# 4. Conclusion

In this study, the physical concepts developed for the theory of phase transitions occurring in bistable systems were for the first time applied to describe the effects of various chemotherapeutic agents on cancer cell lines. Specifically, we adopted the Ising model of a spin system and applied it to the survival plots of cancer cells at different concentrations of the various chemotherapeutic agents these cells were exposed to. This model was originally proposed for a spin system in a uniform external field with a constant interaction parameter and a variable temperature. In the case of cancer cells, the external field is analogous to the logarithm of the drug concentration, while the interaction parameter describes the cell–cell interactions and hence accounts for the bystander effect. Unlike in physical systems, temperature is kept constant for cancer cells in the reported assays. It should be noted that this model has been successfully applied to both interacting and non-interacting cells depending on the underlying biological situation. We have tested the model on a consistently produced dataset of 66 cancer cell lines exposed to 13 different cancer chemotherapy drugs. The results show good agreement between the cell response-Ising model and the biological data. Using the bistabiliy concept in the Ising model, IC50 values can be very accurately determined with an error on the order of 1 nM by one of the parameters of the fitting function in the non-interacting case. The cell–cell interaction was also applied to the experimental data, although in our case, most of the cell lines tend to be non-interacting. Nonetheless, the presence of interactions can be determined using our fitting procedures, and it offers a clear biological insight that bare experimental data do not reveal. We have additionally introduced the thermodynamic concept of the susceptibility function and found its peak to closely coincide with the value of IC50. Further studies should be performed considering a non-constant interaction term as well as non-uniform fields in the Ising model applied to cytotoxicity assays with both two-dimensional and three-dimensional geometries and various cell concentrations. In addition, spatial dimensionality of the cancer cell culture was shown to affect the response to cytotoxic agents, which requires a future study to gain insight into how two-dimensional culture may not be an appropriate proxy for tissue-based studies. This approach is expected to introduce a high level of consistency in cytotoxic data analysis and hence better confidence in the preclinical data assessment for cancer chemotherapy and related applications.

Data accessibility. The experimental IC50 values for all of the studied cell lines exposed to different cytotoxic compounds are provided in the Dryad Digital Repository [https://datadryad.org/stash/share/lJAhCLpM9QSYhmDUn1XP-Sx9sOAKe4rg5QD 5i7Cd2e4], DOI [https://dx.doi.org/10.5061/dryad.4qrfj6q6d] and refs. [42,43,53]. Also, the corresponding scripts for analysing the data have been provided for dose–response curves. In addition, more information has been provided in the electronic supplementary material as well.
Authors' contributions. V.R. and J.A.T. conceived of the project. V.R. and S.A.M. developed the methodology, S.A.M. performed the simulations and all the authors contributed to the writing and editing of the paper.
Competing interests. We declare we have no competing interests.
Funding. This research was supported by an individual Discovery Grant from NSERC (Canada) awarded to J.A.T.
Acknowledgements. We gratefully acknowledge Dr Guido Zaman and his laboratory, Netherlands Translational Research Center B.V. (Oncolines) for providing the experimental data of inhibition profiles of different cell lines. We also thank Drs Zaman and Joost Uitdenhaag for their insightful comments and suggestions and Dr Jordane Preto for his help with the initial testing of the computer code.

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
