## [Reviewer comments · Royal Society Open Science]

Review History

RSOS-191578.R0 (Original submission)

Review form: Reviewer 1

Is the manuscript scientifically sound in its present form?

Yes

Are the interpretations and conclusions justified by the results?

Yes

Is the language acceptable?

Yes

Do you have any ethical concerns with this paper?

No

Have you any concerns about statistical analyses in this paper?

No

Recommendation?

Accept with minor revision (please list in comments)

Comments to the Author(s)

Please see the attachment (Appendix A).

Review form: Reviewer 2

Is the manuscript scientifically sound in its present form?

No

Are the interpretations and conclusions justified by the results?

No

Is the language acceptable?

No

Do you have any ethical concerns with this paper?

No

Have you any concerns about statistical analyses in this paper?

Yes

Recommendation?

Reject

Comments to the Author(s)

Sorry, after delving into this manuscript I realized that I am not capable of translating the physical math into biology.

Decision letter (RSOS-191578.R0)

16-Dec-2019

Dear Dr Tuszynski

On behalf of the Editors, I am pleased to inform you that your Manuscript RSOS-191578 entitled "Cell Death and Survival Due to Cytotoxic Exposure Modeled as a Two-State Ising System" has been accepted for publication in Royal Society Open Science subject to minor revision in accordance with the referee suggestions. Please find the referees' comments at the end of this email.

The Associate Editor has made their recommendation on the basis of the reviewer supplying the attachment, which provides a number of useful suggestions.

The reviewers and handling editors have recommended publication, but also suggest some minor revisions to your manuscript. Therefore, I invite you to respond to the comments and revise your manuscript.

- Ethics statement

- Data accessibility

<http://datadryad.org/submit?journalID=RSOS&manu=RSOS-191578>

- Competing interests

- Authors' contributions

- Acknowledgements

- Funding statement

Because the schedule for publication is very tight, it is a condition of publication that you submit

the revised version of your manuscript before 25-Dec-2019. Please note that the revision deadline will expire at 00.00am on this date. If you do not think you will be able to meet this date please let me know immediately.

If your manuscript is newly submitted and subsequently accepted for publication, you will be asked to pay the article processing charge, unless you request a waiver and this is approved by Royal Society Publishing. You can find out more about the charges at

<https://royalsocietypublishing.org/rsos/charges>. Should you have any queries, please contact openscience@royalsociety.org.

on behalf of Dr Marco Viceconti (Associate Editor) and Pietro Cicuta (Subject Editor)
openscience@royalsociety.org

Reviewer comments to Author:
Reviewer: 1

Comments to the Author(s)
Please see the attachment.

Reviewer: 2

Comments to the Author(s)
Sorry, after delving into this manuscript I realized that I am not capable of translating the physical math into biology.

Author's Response to Decision Letter for (RSOS-191578.R0)

See Appendix B.

Decision letter (RSOS-191578.R1)

08-Jan-2020

Dear Dr Tuszynski,

It is a pleasure to accept your manuscript entitled "Cell Death and Survival Due to Cytotoxic Exposure Modeled as a Two-State Ising System" in its current form for publication in Royal Society Open Science.

Please ensure that you send to the editorial office an editable version of your accepted manuscript, and individual files for each figure and table included in your manuscript. You can send these in a zip folder if more convenient. Failure to provide these files may delay the

processing of your proof. You may disregard this request if you have already provided these files to the editorial office.

Kind regards,
Lianne Parkhouse
Editorial Coordinator
Royal Society Open Science
openscience@royalsociety.org

on behalf of Dr Marco Viceconti (Associate Editor) and Pietro Cicuta (Subject Editor)
openscience@royalsociety.org

Appendix A

Cell Death and Survival Due to Cytotoxic Exposure Modeled as a Two-State Ising System

S. Arbabi Moghadam, V. Rezania and J.A. Tuszynski

The paper provides a novel mathematical description of the cancer cell response to chemotherapeutic agent exposure using a two-state Ising model – a very well-known model of ferromagnetism in statistical mechanics. The cell response-Ising model has shown good agreement with a data set of 66 cancer cell lines exposed to 13 different cancer chemotherapy drugs.

The paper is very well written with few grammatical errors, although I did spot a few very minor issues (accidental spaces before commas in a couple places, inconsistencies in capitalization in figure axis labels, etc.), so I would suggest the authors go through the manuscript and fix these.

The authors have elegantly implemented standard techniques from statistical physics into the modeling of cancer cell survival. As mentioned by the authors in the conclusions section, there are several extensions to this approach. In my opinion, the authors have opened the door to a new perspective on cancer modeling and this work has the potential to have a high impact.

While I have not gone through and checked the math line-by-line, their formulas look like pretty standard statistical mechanics relations. One suggestion I have is that in arriving at Equations (15) and (16), the authors could put more of the calculation details into an appendix.

The references need to be updated as well. For example, the citation 22-43 ranges from Cancer biology book to a number paper on Ising model, and the main reference [5] is not correct (Physica A 416 (2014) 242–251).

Chair

Alexander J.B. (Sandy) McEwan
Tel: 780.432.8320
Fax: 780.432.8425
sandy.mcewan@albertahealthservices.ca

Associate Chair, Graduate Studies

Andrew Shaw
Tel: 780.432.8930
Fax: 780.432.8425
andrew.shaw@albertahealthservices.ca

Assistant Chair, Administration

Cynthia Henderson
Tel: 780.432.8576
Fax: 780.432.8425
cynthia.henderson@albertahealthservices.ca

Director, Experimental Oncology

David Murray
Tel: 780.432.8427
Fax: 780.432.8428
david.murray5@albertahealthservices.ca

Director, Medical Oncology

Peter Venner
Tel: 780.432.8756
Fax: 780.432.8888
peter.venner@albertahealthservices.ca

Director, Medical Physics

B. Gino Fallone
Tel: 780.432.8750
Fax: 780.432.8615
gino.fallone@albertahealthservices.ca

Acting Director, Oncologic Imaging

John Mercer
Tel: 780.989.4311
Fax: 780.432.8483
john.mercer@albertahealthservices.ca

Director, Palliative Care Medicine

Robin Fainsinger
Tel: 780.735.7727
Fax: 780.735.7302
robin.fainsinger@albertahealthservices.ca

Director, Radiation Oncology

Matthew Parliament
Tel: 780.432.8749
Fax: 780.432.8380
matthew.parliament@albertahealthservices.ca

Director, Surgical Oncology

Todd McMullen
Tel: 780.432.8337
Fax: 780.432.8333
todd.mcmullen2@albertahealthservices.ca

December 23, 2019

Dear Journal of Royal Society Open Science Editorial Board,

Please find enclosed our revised manuscript entitled:

**“Cell Death and Survival Due to Cytotoxic Exposure Modeled as a
Two-State Ising System”**

by S. Arbabi Moghadam, V. Rezanian and J. A. Tuszynski. This manuscript is resubmitted as a research article following a recommendation for minor revisions by one of the Journal’s referees. All co-authors have seen and agree to the contents of the manuscript, and we have no conflict of interest to report.

Below, we respond in detail to the referee’s recommendations.

We have made the following changes to the manuscript:

- Transferring equation 2-14 to the SI but keep the explanations for CM (Reviewer's suggestions)
- Adding the following line before Eq.(2): "Using statistical mechanic approaches and mean-field approximation for interacting and non-interacting cells, the partition function, Z, and the order parameter, M are found after several steps of calculations as (see SI Text for calculation details),"
- Renumbering equations according to the changes
- Changing the notation for referring to equations and figures: Eq.(1) to equation (1) and Fig.(1) to figure(1)
- Changing SI Appendix 1 to SI appendix
- Updating the references: References 22-43 were changed to 17,22-42, Reference [22-28] [17-21]
- Ref 31 was removed (non-english paper): W. Heisenberg, Z. Phys. 49, 619 (1928).
- Submitting dataset in another Dryad repository (Due to a mistake on Dr. Rezanian University name)

We hope that these changes properly reflect the reviewer's recommendations and we hope the paper now qualifies for publication in your Journal.

Respectfully,

Prof. Jack Tuszynski

Allard Research Chair
University of Alberta
Division of Experimental Oncology
Room 3336, Cross Cancer Institute
11560 University Avenue
Edmonton, AB T6G 1Z2, Canada

email: jackt@ualberta.ca
Fax: [\(780\) 432-8892](tel:(780)432-8892)
Voice: [\(780\) 432-8906](tel:(780)432-8906)
cell: [\(780\) 964-4517](tel:(780)964-4517)

Professor
Department of Physics
University of Alberta